# Active Learning for Probabilistic Hypotheses Using the Maximum Gibbs Error Criterion

**Nguyen Viet Cuong**      **Wee Sun Lee**      **Nan Ye**
Department of Computer Science
National University of Singapore
{nvcuong,leews,yenan}@comp.nus.edu.sg

**Kian Ming A. Chai**      **Hai Leong Chieu**
DSO National Laboratories, Singapore
{ckianmin,chaileon}@dso.org.sg

## Abstract

We introduce a new objective function for pool-based Bayesian active learning with probabilistic hypotheses. This objective function, called the policy Gibbs error, is the expected error rate of a random classifier drawn from the prior distribution on the examples adaptively selected by the active learning policy. Exact maximization of the policy Gibbs error is hard, so we propose a greedy strategy that maximizes the Gibbs error at each iteration, where the Gibbs error on an instance is the expected error of a random classifier selected from the posterior label distribution on that instance. We apply this maximum Gibbs error criterion to three active learning scenarios: non-adaptive, adaptive, and batch active learning. In each scenario, we prove that the criterion achieves near-maximal policy Gibbs error when constrained to a fixed budget. For practical implementations, we provide approximations to the maximum Gibbs error criterion for Bayesian conditional random fields and transductive Naive Bayes. Our experimental results on a named entity recognition task and a text classification task show that the maximum Gibbs error criterion is an effective active learning criterion for noisy models.

## 1   Introduction

In pool-based active learning [1], we select training data from a finite set (called a pool) of unlabeled examples and aim to obtain good performance on the set by asking for as few labels as possible. If a large enough pool is sampled from the true distribution, good performance of a classifier on the pool implies good generalization performance of the classifier. Previous theoretical works on Bayesian active learning mainly deal with the noiseless case, which assumes a prior distribution on a collection of deterministic mappings from observations to labels [2, 3]. A fixed deterministic mapping is then drawn from the prior, and it is used to label the examples.

In this paper, probabilistic hypotheses, rather than deterministic ones, are used to label the examples. We formulate the objective as a maximum coverage objective with a fixed budget: with a budget of $k$ queries, we aim to select $k$ examples such that the *policy Gibbs error* is maximal. The policy Gibbs error of a policy is the expected error rate of a Gibbs classifier[1] on the set adaptively selected by the policy. The policy Gibbs error is a lower bound of the policy entropy, a generalization of the Shannon entropy to general (both adaptive and non-adaptive) policies. For non-adaptive policies,

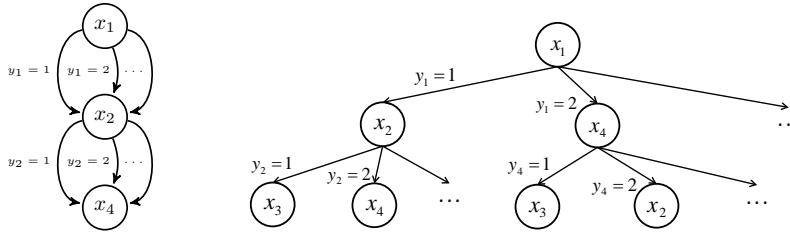

Figure 1: An example of a non-adaptive policy tree (left) and an adaptive policy tree (right).

the policy Gibbs error reduces to the Gibbs error for sets, which is a special case of a measure of uncertainty called the Tsallis entropy [4].

By maximizing policy Gibbs error, we hope to maximize the policy entropy, whose maximality implies the minimality of the posterior label entropy of the remaining unlabeled examples in the pool. Besides, by maximizing policy Gibbs error, we also aim to obtain a small expected error of a posterior Gibbs classifier (which samples a hypothesis from the posterior instead of the prior for labeling). Small expected error of the posterior Gibbs classifier is desirable as it upper bounds the Bayes error but is at most twice of it.

Maximizing policy Gibbs error is hard, and we propose a greedy criterion, the *maximum Gibbs error criterion* (maxGEC), to solve it. With this criterion, the next query is made on the candidate (which may be one or several examples) that has maximum Gibbs error, the probability that a randomly sampled labeling does not match the actual labeling. We investigate this criterion in three settings: the non-adaptive setting, the adaptive setting and batch setting (also called batch mode setting) [5]. In the non-adaptive setting, the set of examples is not labeled until all examples in the set have all been selected. In the adaptive setting, the examples are labeled as soon as they are selected, and the new information is used to select the next example. In the batch setting, we select a batch of examples, query their labels and proceed to select the next batch taking into account the labels. In all these settings, we prove that maxGEC is near-optimal compared to the best policy that has maximal policy Gibbs error in the setting.

We examine how to compute the maxGEC criterion, particularly for large structured probabilistic models such as the conditional random fields [6]. When inference in the conditional random field can be done efficiently, we show how to compute an approximation to the Gibbs error by sampling and efficient inference. We provide an approximation for maxGEC in the non-adaptive and batch settings with Bayesian transductive Naive Bayes model. Finally, we conduct pool-based active learning experiments using maxGEC for a named entity recognition task with conditional random fields and a text classification task with Bayesian transductive Naive Bayes. The results show good performance of maxGEC in terms of the area under the curve (AUC).

## 2  Preliminaries

Let $\mathcal{X}$ be a set of examples, $\mathcal{Y}$ be a fixed finite set of labels, and $\mathcal{H}$ be a set of probabilistic hypotheses. We assume $\mathcal{H}$ is finite, but our results extend readily to general $\mathcal{H}$. For any probabilistic hypothesis $h \in \mathcal{H}$, its application to an example $x \in \mathcal{X}$ is a categorical random variable with support $\mathcal{Y}$, and we write $\mathbb{P}[h(x) = y|h]$ for the probability that $h(x)$ has value $y \in \mathcal{Y}$. We extend the notation to any sequence $S$ of examples from $\mathcal{X}$ and write $\mathbb{P}[h(S) = \mathbf{y}|h]$ for the probability that $h(S)$ has a labeling $\mathbf{y} \in \mathcal{Y}^{|S|}$, where $\mathcal{Y}^{|S|}$ is the set of all labelings of $S$. We operate within the Bayesian setting and assume a prior probability $p_0[h]$ on $\mathcal{H}$. We use $p_{\mathcal{D}}[h]$ to denote the posterior $p_0[h|\mathcal{D}]$ after observing a set $\mathcal{D}$ of labeled examples from $\mathcal{X} \times \mathcal{Y}$.

A pool-based active learning algorithm is a policy for choosing training examples from a pool $X \subseteq \mathcal{X}$. At the beginning, a fixed labeling $\mathbf{y}^*$ of $X$ is given by a hypothesis $h$ drawn from the prior $p_0[h]$ and is hidden from the learner. Equivalently, $\mathbf{y}^*$ can be drawn from the prior label distribution $p_0[\mathbf{y}^*; X]$. For any distribution $p[h]$, we use $p[\mathbf{y}; S]$ to denote the probability that examples in $S$ are assigned the labeling $\mathbf{y}$ by a hypothesis drawn randomly from $p[h]$. Formally, $p[\mathbf{y}; S] \stackrel{\text{def}}{=} \sum_{h \in \mathcal{H}} p[h] \mathbb{P}[h(S) = \mathbf{y}|h]$. When $S$ is a singleton $\{x\}$, we write $p[y; x]$ for $p[\{y\}; \{x\}]$.

During the learning process, each time the learner selects an unlabeled example, its label will be revealed to the learner. A policy for choosing training examples is a mapping from a set of labeled examples to an unlabeled example to be queried. This can be represented by a policy tree, where a node represents the next example to be queried, and each edge from the node corresponds to a possible label. We use policy and policy tree as synonyms. Figure 1 illustrates two policy trees with their top three levels: in the non-adaptive setting, the policy ignores the labels of the previously selected examples, so all examples at the same depth of the policy tree are the same; in the adaptive setting, the policy takes into account the observed labels when choosing the next example.

A full policy tree for a pool $X$ is a policy tree of height $|X|$. A partial policy tree is a subtree of a full policy tree with the same root. The class of policies of height $k$ will be denoted by $\Pi_k$. Our query criterion gives a method to build a full policy tree one level at a time. The main building block is the probability distribution $p_0^\pi[\cdot]$ over all possible paths from the root to the leaves for any (full or partial) policy tree $\pi$. This distribution over paths is induced from the uncertainty in the fixed labeling $\mathbf{y}^*$ for $X$: since $\mathbf{y}^*$ is drawn randomly from $p_0[\mathbf{y}^*; X]$, the path $\rho$ followed from the root to a leaf of the policy tree during the execution of $\pi$ is also a random variable. If $x_\rho$ (resp. $y_\rho$) is the sequence of examples (resp. labels) along path $\rho$, then the probability of $\rho$ is $p_0^\pi[\rho] \stackrel{\text{def}}{=} p_0[y_\rho; x_\rho]$.

## 3 Maximum Gibbs Error Criterion for Active Learning

A commonly used objective for active learning in the non-adaptive setting is to choose $k$ training examples such that their Shannon entropy is maximal, as this reduces uncertainty in the later stage. We first give a generalization of the concept of Shannon entropy to general (both adaptive and non-adaptive) policies. Formally, the policy entropy of a policy $\pi$ is

$$H(\pi) \stackrel{\text{def}}{=} \mathbb{E}_{\rho \sim p_0^\pi}\big[ -\ln p_0^\pi[\rho] \big].$$

From this definition, policy entropy is the Shannon entropy of the paths in the policy. The policy entropy reduces to the Shannon entropy on a set of examples when the policy is non-adaptive. The following result gives a formal statement that maximizing policy entropy minimizes the uncertainty on the label of the remaining unlabeled examples in the pool. Suppose a path $\rho$ has been observed, the labels of the remaining examples in $X \setminus x_\rho$ follow the distribution $p_\rho[\cdot; X \setminus x_\rho]$, where $p_\rho$ is the posterior obtained after observing $(x_\rho, y_\rho)$. The entropy of this distribution will be denoted by $G(\rho)$ and will be called the posterior label entropy of the remaining examples given $\rho$. Formally, $G(\rho) = -\sum_{\mathbf{y}} p_\rho[\mathbf{y}; X \setminus x_\rho] \ln p_\rho[\mathbf{y}; X \setminus x_\rho]$, where the summation is over all possible labelings $\mathbf{y}$ of $X \setminus x_\rho$. The posterior label entropy of a policy $\pi$ is defined as $G(\pi) = \mathbb{E}_{\rho \sim p_0^\pi} G(\rho)$.

**Theorem 1.** *For any $k \geq 1$, if a policy $\pi$ in $\Pi_k$ maximizes $H(\pi)$, then $\pi$ minimizes the posterior label entropy $G(\pi)$.*

*Proof.* It can be easily verified that $H(\pi) + G(\pi)$ is the Shannon entropy of the label distribution $p_0[\cdot; X]$, which is a constant (detailed proof is in the supplementary). Thus, the theorem follows. $\square$

The usual maximum Shannon entropy criterion, which selects the next example $x$ maximizing $\mathbb{E}_{y \sim p_\mathcal{D}[y;x]}[-\ln p_\mathcal{D}[y;x]]$ where $\mathcal{D}$ is the previously observed labeled examples, can be thought of as a greedy heuristic for building a policy $\pi$ maximizing $H(\pi)$. However, it is still unknown whether this greedy criterion has any theoretical guarantee, except for the non-adaptive case.

In this paper, we introduce a new objective for active learning: the policy Gibbs error. This new objective is a lower bound of the policy entropy and there are near-optimal greedy algorithms to optimize it. Intuitively, the policy Gibbs error of a policy $\pi$ is the expected probability for a Gibbs classifier to make an error on the set adaptively selected by $\pi$. Formally, we define the *policy Gibbs error* of a policy $\pi$ as

$$V(\pi) \stackrel{\text{def}}{=} \mathbb{E}_{\rho \sim p_0^\pi}\big[ 1 - p_0^\pi[\rho] \big], \tag{1}$$

In the above equation, $1 - p_0^\pi[\rho]$ is the probability that a Gibbs classifier makes an error on the selected set along the path $\rho$. Theorem 2 below, which is straightforward from the inequality $x \geq 1 + \ln x$, states that the policy Gibbs error is a lower bound of the policy entropy.

**Theorem 2.** *For any (full or partial) policy $\pi$, we have $V(\pi) \leq H(\pi)$.*

Given a budget of $k$ queries, our proposed objective is to find $\pi^* = \arg\max_{\pi \in \Pi_k} V(\pi)$, the height $k$ policy with maximum policy Gibbs error. By maximizing $V(\pi)$, we hope to maximize the policy entropy $H(\pi)$, and thus minimize the uncertainty in the remaining examples. Furthermore, we also hope to obtain a small expected error of a posterior Gibbs classifier, which upper bounds the Bayes error but is at most twice of it. Using this objective, we propose greedy algorithms for active learning that are provably near-optimal for probabilistic hypotheses. We will consider the non-adaptive, adaptive and batch settings.

## 3.1 The Non-adaptive Setting

In the non-adaptive setting, the policy $\pi$ ignores the observed labels: it never updates the posterior. This is equivalent to selecting a set of examples before any labeling is done. In this setting, the examples selected along all paths of $\pi$ are the same. Let $x_\pi$ be the set of examples selected by $\pi$. The Gibbs error of a non-adaptive policy $\pi$ is simply

$$V(\pi) = \mathbb{E}_{\mathbf{y} \sim p_0[\,\cdot\,; x_\pi]}[1 - p_0[\mathbf{y}; x_\pi]].$$

Thus, the optimal non-adaptive policy selects a set $S$ of examples maximizing its Gibbs error, which is defined by $\epsilon_g^{p_0}(S) \stackrel{\text{def}}{=} 1 - \sum_{\mathbf{y}} p_0[\mathbf{y}; S]^2$.

In general, the Gibbs error of a distribution $P$ is $1 - \sum_i P[i]^2$, where the summation is over elements in the support of $P$. The Gibbs error is a special case of the Tsallis entropy used in nonextensive statistical mechanics [4] and is known to be monotone submodular [7]. From the properties of monotone submodular functions [8], the greedy non-adaptive policy that selects the next example

$$x_{i+1} = \arg\max_x \{\epsilon_g^{p_0}(S_i \cup \{x\})\} = \arg\max_x \{1 - \sum_{\mathbf{y}} p_0[\mathbf{y}; S_i \cup \{x\}]^2\}, \qquad (2)$$

where $S_i$ is the set of previously selected examples, is near-optimal compared to the best non-adaptive policy. This is stated below.

**Theorem 3.** *Given a budget of $k \geq 1$ queries, let $\pi_n$ be the non-adaptive policy in $\Pi_k$ selecting examples using Equation* (2), *and let $\pi_n^*$ be the non-adaptive policy in $\Pi_k$ with the maximum policy Gibbs error. Then, $V(\pi_n) > (1 - 1/e)V(\pi_n^*)$.*

## 3.2 The Adaptive Setting

In the adaptive setting, a policy takes into account the observed labels when choosing the next example. This is done via the posterior update after observing the label of a selected example. The adaptive setting is the most common setting for active learning. We now describe a greedy adaptive algorithm for this setting that is near-optimal. Assume that the current posterior obtained after observing the labeled examples $\mathcal{D}$ is $p_\mathcal{D}$. Our greedy algorithm selects the next example $x$ that maximizes $\epsilon_g^{p_\mathcal{D}}(x)$:

$$x^* = \arg\max_x \epsilon_g^{p_\mathcal{D}}(x) = \arg\max_x \{1 - \sum_{y \in \mathcal{Y}} p_\mathcal{D}[y; x]^2\}. \qquad (3)$$

From the definition of $\epsilon_g^{p_\mathcal{D}}$ in Section 3.1, $\epsilon_g^{p_\mathcal{D}}(x)$ is in fact the Gibbs error of a 1-step policy with respect to the prior $p_\mathcal{D}$. Thus, we call this greedy criterion the adaptive *maximum Gibbs error criterion* (maxGEC). Note that in binary classification where $|\mathcal{Y}| = 2$, maxGEC selects the same example as the maximum Shannon entropy and the least confidence criteria. However, they are different in the multi-class case. Theorem 4 below states that maxGEC is near-optimal compared to the best adaptive policy with respect to the objective in Equation (1).

**Theorem 4.** *Given a budget of $k \geq 1$ queries, let $\pi^{\text{maxGEC}}$ be the adaptive policy in $\Pi_k$ selecting examples using* maxGEC *and $\pi^*$ be the adaptive policy in $\Pi_k$ with the maximum policy Gibbs error. Then, $V(\pi^{\text{maxGEC}}) > (1 - 1/e)V(\pi^*)$.*

The proof for this theorem is in the supplementary material. The main idea of the proof is to reduce probabilistic hypotheses to deterministic ones by expanding the hypothesis space. For deterministic hypotheses, we show that maxGEC is equivalent to maximizing the version space reduction objective, which is known to be adaptive monotone submodular [2]. Thus, we can apply a known result for optimizing adaptive monotone submodular function [2] to obtain Theorem 4.

---
**Algorithm 1** Batch maxGEC for Bayesian Batch Active Learning
---
 **Input:** Unlabeled pool $X$, prior $p_0$, number of iterations $k$, and batch size $s$.
 **for** $i = 0$ **to** $k - 1$ **do**
   $S \leftarrow \emptyset$
   **for** $j = 0$ **to** $s - 1$ **do**
     $x^* \leftarrow \arg \max_x \epsilon_g^{p_i}(S \cup \{x\}); \quad S \leftarrow S \cup \{x^*\}; \quad X \leftarrow X \setminus \{x^*\}$
   **end for**
   $y_S \leftarrow$ Query-labels$(S); \quad p_{i+1} \leftarrow$ Posterior-update$(p_i, S, y_S)$
 **end for**
---

## 3.3 The Batch Setting

In the batch setting [5], we query the labels of $s$ (instead of 1) examples each time, and we do this for a given number of $k$ iterations. After each iteration, we query the labeling of the selected batch and update the posterior based on this labeling. The new posterior can be used to select the next batch of examples. A non-adaptive policy can be seen as a batch policy that selects only one batch. Algorithm 1 describes a greedy algorithm for this setting which we call the *batch maxGEC* algorithm. At iteration $i$ of the algorithm with the posterior $p_i$, the batch $S$ is first initialized to be empty, then $s$ examples are greedily chosen one at a time using the criterion

$$x^* = \arg \max_x \epsilon_g^{p_i}(S \cup \{x\}). \tag{4}$$

This is equivalent to running the non-adaptive greedy algorithm in Section 3.1 to select each batch. Query-labels$(S)$ returns the true labeling $y_S$ of $S$ and Posterior-update$(p_i, S, y_S)$ returns the new posterior obtained from the prior $p_i$ after observing $y_S$.

The following theorem states that batch maxGEC is near optimal compared to the best batch policy with respect to the objective in Equation (1). The proof for this theorem is in the supplementary material. The proof also makes use of the reduction to deterministic hypotheses and the adaptive submodularity of version space reduction.

**Theorem 5.** *Given a budget of $k$ batches of size $s$, let $\pi_b^{\mathrm{maxGEC}}$ be the batch policy selecting $k$ batches using* batch maxGEC *and $\pi_b^*$ be the batch policy selecting $k$ batches with maximum policy Gibbs error. Then, $V(\pi_b^{\mathrm{maxGEC}}) > (1 - e^{-(e-1)/e})V(\pi_b^*)$.*

This theorem has a different bounding constant than those in Theorems 3 and 4 because it uses two levels of approximation to compute the batch policy: at each iteration, it approximates the optimal batch by greedily choosing one example at a time using equation (4) ($1^{st}$ approximation). Then it uses these chosen batches to approximate the optimal batch policy ($2^{nd}$ approximation). In contrast, the fully adaptive case has batch size 1 and only needs the $2^{nd}$ approximation, while the non-adaptive case chooses 1 batch and only needs the $1^{st}$ approximation.

In non-adaptive and batch settings, our algorithms need to sum over all labelings of the previously selected examples in a batch to choose the next example. This summation is usually expensive and it restricts the algorithms to small batches. However, we note that small batches may be preferred in some real problems. For example, if there is a small number of annotators and labeling one example takes a long time, we may want to select a batch size that matches the number of annotators. In this case, the annotators can label the examples concurrently while we can make use of the labels as soon as they are available. It would take a longer time to label a larger batch and we cannot use the labels until all the examples in the batch are labeled.

## 4 Computing maxGEC

We now discuss how to compute maxGEC and batch maxGEC for some probabilistic models. Computing the values is often difficult and we discuss some sampling methods for this task.

### 4.1 MaxGEC for Bayesian Conditional Exponential Models

A conditional exponential model defines the conditional probability $P_\lambda[\vec{y}|\vec{x}]$ of a structured labels $\vec{y}$ given a structured inputs $\vec{x}$ as $P_\lambda[\vec{y}|\vec{x}] = \exp\left(\sum_{i=1}^m \lambda_i F_i(\vec{y}, \vec{x})\right)/Z_\lambda(\vec{x})$, where $\lambda =$

**Algorithm 2** Approximation for Equation (4).

---
**Input:** Selected unlabeled examples $S$, current unlabeled example $x$, current posterior $p_{\mathcal{D}}^c$.
Sample $M$ label vectors $(\mathbf{y}^i)_{i=0}^{M-1}$ of $(X \setminus T) \cup \mathcal{T}$ from $p_{\mathcal{D}}^c$ using Gibbs sampling and set $r \leftarrow 0$.
**for** $i = 0$ **to** $M - 1$ **do**
  **for** $y \in \mathcal{Y}$ **do**
    $\widehat{p_{\mathcal{D}}^c}[h(S) = \mathbf{y}_S^i \wedge h(x) = y] \leftarrow M^{-1} \left| \left\{ \mathbf{y}^j \mid \mathbf{y}_S^j = \mathbf{y}_S^i \wedge \mathbf{y}_{\{x\}}^j = y \right\} \right|$
    $r \leftarrow r + (\widehat{p_{\mathcal{D}}^c}[h(S) = \mathbf{y}_S^i \wedge h(x) = y])^2$
  **end for**
**end for**
**return** $1 - r$

---

$(\lambda_1, \lambda_2, \ldots, \lambda_m)$ is the parameter vector, $F_i(\vec{y}, \vec{x})$ is the total score of the $i$-th feature, and $Z_\lambda(\vec{x}) = \sum_{\vec{y}} \exp\left(\sum_{i=1}^m \lambda_i F_i(\vec{y}, \vec{x})\right)$ is the partition function. A well-known conditional exponential model is the linear-chain conditional random field (CRF) [6] in which $\vec{x}$ and $\vec{y}$ both have sequence structures. That is, $\vec{x} = (x_1, x_2, \ldots, x_{|\vec{x}|}) \in \mathcal{X}^{|\vec{x}|}$ and $\vec{y} = (y_1, y_2, \ldots, y_{|\vec{x}|}) \in \mathcal{Y}^{|\vec{x}|}$. In this model, $F_i(\vec{y}, \vec{x}) = \sum_{j=1}^{|\vec{x}|} f_i(y_j, y_{j-1}, \vec{x})$ where $f_i(y_j, y_{j-1}, \vec{x})$ is the score of the $i$-th feature at position $j$.

In the Bayesian setting, we assume a prior $p_0[\lambda] = \prod_{i=1}^m p_0[\lambda_i]$ on $\lambda$, where $p_0[\lambda_i] = \mathcal{N}(\lambda_i|0, \sigma^2)$ for a known $\sigma$. After observing the labeled examples $\mathcal{D} = \{(\vec{x}_j, \vec{y}_j)\}_{j=1}^t$, we can obtain the posterior

$$p_{\mathcal{D}}[\lambda] = p_0[\lambda|\mathcal{D}] \propto \prod_{j=1}^t \frac{1}{Z_\lambda(\vec{x}_j)} \exp\left(\sum_{i=1}^m \lambda_i F_i(\vec{y}_j, \vec{x}_j)\right) \exp\left(-\frac{1}{2}\sum_{i=1}^m \left(\frac{\lambda_i}{\sigma}\right)^2\right).$$

For active learning, we need to estimate the Gibbs error in Equation (3) from the posterior $p_{\mathcal{D}}$. For each $\vec{x}$, we can approximate the Gibbs error $\epsilon_g^{p_{\mathcal{D}}}(\vec{x}) = 1 - \sum_{\vec{y}} p_{\mathcal{D}}[\vec{y}; \vec{x}]^2$ by sampling $N$ hypotheses $\lambda^1, \lambda^2, \ldots, \lambda^N$ from the posterior $p_{\mathcal{D}}$. In this case, $\epsilon_g^{p_{\mathcal{D}}}(\vec{x}) \approx 1 - N^{-2} \sum_{j=1}^N \sum_{t=1}^N Z_{\lambda^j + \lambda^t}(\vec{x})/Z_{\lambda^j}(\vec{x})Z_{\lambda^t}(\vec{x})$. The derivation for this formula is in the supplementary material. If we only use the MAP hypothesis $\lambda^*$ to approximate the Gibbs error (i.e. the non-Bayesian setting), then $N = 1$ and $\epsilon_g^{p_{\mathcal{D}}}(\vec{x}) \approx 1 - Z_{2\lambda^*}(\vec{x})/Z_{\lambda^*}(\vec{x})^2$.

This approximation can be done efficiently if we can compute the partition functions $Z_\lambda(\vec{x})$ efficiently for any $\lambda$. This condition holds for a wide range of models including logistic regression, linear-chain CRF, semi-Markov CRF [9], and sparse high-order semi-Markov CRF [10].

## 4.2 Batch maxGEC for Bayesian Transductive Naive Bayes

We discuss an algorithm to approximate batch maxGEC for non-adaptive and batch active learning with Bayesian transductive Naive Bayes. First, we describe the Bayesian transductive Naive Bayes model for text classification. Let $Y \in \mathcal{Y}$ be a random variable denoting the label of a document and $W \in \mathcal{W}$ be a random variable denoting a word. In a Naive Bayes model, the parameters are $\theta = \{\theta_y\}_{y \in \mathcal{Y}} \cup \{\theta_{w|y}\}_{w \in \mathcal{W}, y \in \mathcal{Y}}$, where $\theta_y = \mathbb{P}[Y = y]$ and $\theta_{w|y} = \mathbb{P}[W = w|Y = y]$. For a document $X$ and a label $Y$, if $X = \{W_1, W_2, \ldots, W_{|X|}\}$ where $W_i$ is a word in the document, we model the joint distribution $\mathbb{P}[X, Y] = \theta_Y \prod_{i=1}^{|X|} \theta_{W_i|Y}$.

In the Bayesian setting, we have a prior $p_0[\theta]$ such that $\theta_y \sim \text{Dirichlet}(\alpha)$ and $\theta_{w|y} \sim \text{Dirichlet}(\alpha_y)$ for each $y$. When we observe the labeled documents, we update the posterior by counting the labels and the words in each document label. The posterior parameters also follow Dirichlet distributions. Let $X$ be the original pool of training examples and $\mathcal{T}$ be the unlabeled testing examples. In transductive setting, we work with the conditional prior $p_0^c[\theta] = p_0[\theta|X; \mathcal{T}]$. For a set $\mathcal{D} = (T, \mathbf{y}_T)$ of labeled examples where $T \subseteq X$ is the set of unlabeled examples and $\mathbf{y}_T$ is the labeling of $T$, the conditional posterior is $p_{\mathcal{D}}^c[\theta] = p_0[\theta|X; \mathcal{T}; \mathcal{D}] = p_{\mathcal{D}}[\theta|(X \setminus T) \cup \mathcal{T}]$, where $p_{\mathcal{D}}[\theta] = p_0[\theta|\mathcal{D}]$ is the Dirichlet posterior of the non-transductive model. To implement the batch maxGEC algorithm, we need to estimate the Gibbs error in Equation (4) from the conditional posterior. Let $S$ be the currently selected batch. For each unlabeled example $x \notin S$, we need to estimate:

$$1 - \sum_{\mathbf{y}_S, y} \left(p_{\mathcal{D}}^c\left[h(S) = \mathbf{y}_S \wedge h(x) = y\right]\right)^2 = 1 - \mathbb{E}_{\mathbf{y}_S}\left[\frac{\sum_y \left(p_{\mathcal{D}}^c\left[h(S) = \mathbf{y}_S \wedge h(x) = y\right]\right)^2}{p_{\mathcal{D}}^c[\mathbf{y}_S; S]}\right],$$

Table 1: AUC of different learning algorithms with batch size $s = 10$.

| Task | TPass | maxGEC | LC | NPass | LogPass | LogFisher |
|---|---|---|---|---|---|---|
| alt.atheism/comp.graphics | 87.43 | 91.69 | 91.66 | 84.98 | 91.63 | **93.92** |
| talk.politics.guns/talk.politics.mideast | 84.92 | 92.03 | **92.16** | 80.80 | 86.07 | 88.36 |
| comp.sys.mac.hardware/comp.windows.x | 73.17 | **93.60** | 92.27 | 74.41 | 85.87 | 88.71 |
| rec.motorcycles/rec.sport.baseball | 93.82 | **96.40** | 96.23 | 92.33 | 89.46 | 93.90 |
| sci.crypt/sci.electronics | 60.46 | 85.51 | 85.86 | 60.85 | 82.89 | **87.72** |
| sci.space/soc.religion.christian | 92.38 | **95.83** | 95.45 | 89.72 | 91.16 | 94.04 |
| soc.religion.christian/talk.politics.guns | 91.57 | **95.94** | 95.59 | 85.56 | 90.35 | 93.96 |
| Average | 83.39 | **93.00** | 92.75 | 81.24 | 88.21 | 91.52 |

where the expectation is with respect to the distribution $p_{\mathcal{D}}^c[\mathbf{y}_S; S]$. We can use Gibbs sampling to approximate this expectation. First, we sample $M$ label vectors $\mathbf{y}_{(X \setminus T) \cup \mathcal{T}}$ of the remaining unlabeled examples from $p_{\mathcal{D}}^c$ using Gibbs sampling. Then, for each $\mathbf{y}_S$, we estimate $p_{\mathcal{D}}^c[\mathbf{y}_S; S]$ by counting the fraction of the $M$ sampled vectors consistent with $\mathbf{y}_S$. For each $\mathbf{y}_S$ and $y$, we also estimate $p_{\mathcal{D}}^c[h(S) = \mathbf{y}_S \wedge h(x) = y]$ by counting the fraction of the $M$ sampled vectors consistent with both $\mathbf{y}_S$ and $y$ on $S \cup \{x\}$. This approximation is equivalent to Algorithm 2. In the algorithm, $\mathbf{y}_S^i$ is the labeling of $S$ according to $\mathbf{y}^i$.

## 5 Experiments

### 5.1 Named Entity Recognition (NER) with CRF

In this experiment, we consider the NER task with the Bayesian CRF model described in Section 4.1. We use a subset of the CoNLL 2003 NER task [11] which contains 1928 training and 969 test sentences. Following the setting in [12], we let the cost of querying the label sequence of each sentence be 1. We implement two versions of maxGEC with the approximation algorithm in Section 4.1: the first version approximates Gibbs error by using only the MAP hypothesis (maxGEC-MAP) and the second version approximates Gibbs error by using 50 hypotheses sampled from the posterior (maxGEC-50). We sample the hypotheses for maxGEC-50 from the posterior by Metropolis-Hastings algorithm with the MAP hypothesis as the initial point.

We compare the maxGEC algorithms with 4 other learning criteria: passive learner (Passive), active learner which chooses the longest unlabeled sequence (Longest), active learner which chooses the unlabeled sequence with maximum Shannon entropy (SegEnt), and active learner which chooses the unlabeled sequence with the least confidence (LeastConf). For SegEnt and LeastConf, the entropy and confidence are estimated from the MAP hypothesis. For all the algorithms, we use the MAP hypothesis for Viterbi decoding. To our knowledge, there is no simple way to compute SegEnt or LeastConf criteria from a finite sample of hypotheses except for using only the MAP estimation. The difficulty is to compute a summation (minimization for LeastConf) over all the outputs $\vec{y}$ in the complex structured models. For maxGEC, the summation can be rearranged to obtain the partition functions, which can be computed efficiently using known inference algorithms. This is thus an advantage of using maxGEC.

We compare the total area under the $F_1$ curve (AUC) for each algorithm after querying the first 500 sentences. As a percentage of the maximum score of 500, algorithms Passive, Longest, SegEnt, LeastConf, maxGEC-MAP and maxGEC-50 attain 72.8, 67.0, 75.4, 75.5, 75.8 and 76.0 respectively. Hence, the maxGEC algorithms perform better than all the other algorithms, and significantly so over the Passive and Longest algorithms.

### 5.2 Text Classification with Bayesian Transductive Naive Bayes

In this experiment, we consider the text classification model in Section 4.2 with the meta-parameters $\alpha = (0.1, \ldots, 0.1)$ and $\alpha_y = (0.1, \ldots, 0.1)$ for all $y$. We implement batch maxGEC (maxGEC) with the approximation in Algorithm 2 and compare with 5 other algorithms: passive learner with Bayesian transductive Naive Bayes model (TPass), least confidence active learner with Bayesian transductive Naive Bayes model (LC), passive learner with Bayesian non-transductive Naive Bayes model (NPass), passive learner with logistic regression model (LogPass), and batch active learner

with Fisher information matrix and logistic regression model (LogFisher) [5]. To implement the least confidence algorithm, we sample $M$ label vectors as in Algorithm 2 and use them to estimate the label distribution for each unlabeled example. The algorithm will then select $s$ examples whose label is least confident according to these estimates.

We run the algorithms on 7 binary tasks from the 20Newsgroups dataset [13] with batch size $s = 10, 20, 30$ and report the areas under the accuracy curve (AUC) for the case $s = 10$ in Table 1. The results for $s = 20, 30$ are in the supplementary material. The results are obtained by averaging over 5 different runs of the algorithms, and the AUCs are normalized so that their range is from 0 to 100. From the results, maxGEC obtains the best AUC scores on 4/7 tasks for each batch size and also the best average AUC scores. LC also performs well and its scores are only slightly lower than maxGEC. The passive learning algorithms are much worse than the active learning algorithms.

## 6  Related Work

Among pool-based active learning algorithms, greedy methods are the simplest and most common [14]. Often, the greedy algorithms try to maximize the uncertainty, e.g. Shannon entropy, of the example to be queried [12]. For non-adaptive active learning, greedy optimization of the Shannon entropy guarantees near optimal performance due to the submodularity of the entropy [2]. However, this has not been shown to extend to adaptive active learning, where each example is labeled as soon as it is selected, and the labeled examples are exploited in selecting the next example to label.

Although greedy algorithms work well in practice [12, 14], they usually do not have any theoretical guarantee except for the case where data are noiseless. In noiseless Bayesian setting, an algorithm called generalized binary search was proven to be near-optimal: its expected number of queries is within a factor of $(\ln \frac{1}{\min_h p_0[h]} + 1)$ of the optimum, where $p_0$ is the prior [2]. This result was obtained using the adaptive submodularity of the version space reduction. Adaptive submodularity is an adaptive version of submodularity, a natural diminishing returns property. The adaptive submodularity of version space reduction was also applied to the batch setting to prove the near-optimality of a batch greedy algorithm that maximizes the average version space reduction for each selected batch [3]. The maxGEC and batch maxGEC algorithms that we proposed in this paper can be seen as generalizations of these version space reduction algorithms to the noisy setting. When the hypotheses are deterministic, our algorithms are equivalent to these version space reduction algorithms.

For the case of noisy data, a noisy version of the generalized binary search was proposed [15]. The algorithm was proven to be optimal under the neighborly condition, a very limited setting where "each hypothesis is locally distinguishable from all others" [15]. In another work, Bayesian active learning was modeled by the Equivalance Class Determination problem and a greedy algorithm called $EC^2$ was proposed for this problem [16]. Although the cost of $EC^2$ is provably near-optimal, this formulation requires an explicit noise model and the near-optimality bound is only useful when the support of the noise model is small. Our formulation, in contrast, is simpler and does not require an explicit noise model: the noise model is implicit in the probabilistic model and our algorithms are only limited by computational concerns.

## 7  Conclusion

We considered a new objective function for Bayesian active learning: the policy Gibbs error. With this objective, we described the maximum Gibbs error criterion for selecting the examples. The algorithm has near-optimality guarantees in the non-adaptive, adaptive and batch settings. We discussed algorithms to approximate the Gibbs error criterion for Bayesian CRF and Bayesian transductive Naive Bayes. We also showed that the criterion is useful for NER with CRF model and for text classification with Bayesian transductive Naive Bayes model.

### Acknowledgments

This work is supported by DSO grant DSOL11102 and the US Air Force Research Laboratory under agreement number FA2386-12-1-4031.

## Footnotes

[1] A Gibbs classifier samples a hypothesis from the prior for labeling.

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
