[Supplementary Material · supplementary.pdf]

# Active Learning for Probabilistic Hypotheses Using the Maximum Gibbs Error Criterion
## — Supplementary Material —

**Nguyen Viet Cuong**     **Wee Sun Lee**     **Nan Ye**
Department of Computer Science
National University of Singapore
{nvcuong,leews,yenan}@comp.nus.edu.sg

**Kian Ming A. Chai**     **Hai Leong Chieu**
DSO National Laboratories, Singapore
{ckianmin,chaileon}@dso.org.sg

## A Detailed Proof of Theorem 1

In the following, let $\rho$ take range as the set of paths from the root to the leaves in the policy $\pi$. The notation $p_0[\mathbf{y}; S]$ means the probability that examples in $S$ are assigned labels $\mathbf{y}$, and we also use $p_0[(\mathbf{y}, \mathbf{y}'); (S, S')]$ to refer to the probability that examples in $S$ and $S'$ are assigned labels $\mathbf{y}$ and $\mathbf{y}'$ respectively. Let $\mathbf{1}(A)$ be the indicator function for the event $A$. In this proof, note that if we fix a labeling $\mathbf{y}$ of $X$, the path $\rho$ followed from the root to a leaf of the policy tree during the execution of the policy $\pi$ is unique (we only consider deterministic policies). The entropy of the distribution $p_0[\,\cdot\,; X]$ is

$$
\begin{aligned}
& -\sum_{\mathbf{y}} p_0[\mathbf{y}; X] \ln p_0[\mathbf{y}; X] \\
= \ & -\sum_{\mathbf{y}}[\sum_{\rho} \mathbf{1}(\mathbf{y} \text{ is consistent with } \rho) p_0[\mathbf{y}; X] \ln p_0[\mathbf{y}; X]] \\
= \ & -\sum_{\rho}[\sum_{\mathbf{y}} \mathbf{1}(\mathbf{y} \text{ is consistent with } \rho) p_0[\mathbf{y}; X] \ln p_0[\mathbf{y}; X]] \\
= \ & -\sum_{\rho}[\sum_{\mathbf{y}'} p_0[(y_\rho, \mathbf{y}'); (x_\rho, X \setminus x_\rho)] \ln p_0[(y_\rho, \mathbf{y}'); (x_\rho, X \setminus x_\rho)]] \\
= \ & -\sum_{\rho}[\sum_{\mathbf{y}'} p_0[(y_\rho, \mathbf{y}'); (x_\rho, X \setminus x_\rho)][\ln p_0[y_\rho; x_\rho] + \ln p_\rho[\mathbf{y}'; X \setminus x_\rho]] \\
= \ & -\sum_{\rho}[\sum_{\mathbf{y}'} p_0[(y_\rho, \mathbf{y}'); (x_\rho, X \setminus x_\rho)] \ln p_0[y_\rho; x_\rho]] \\
& -\sum_{\rho}[\sum_{\mathbf{y}'} p_0[(y_\rho, \mathbf{y}'); (x_\rho, X \setminus x_\rho)] \ln p_\rho[\mathbf{y}'; X \setminus x_\rho]] \\
= \ & -\sum_{\rho} p_0[y_\rho; x_\rho] \ln p_0[y_\rho; x_\rho] - \sum_{\rho}[\sum_{\mathbf{y}'} p_0[y_\rho; x_\rho] p_\rho[\mathbf{y}'; X \setminus x_\rho] \ln p_\rho[\mathbf{y}'; X \setminus x_\rho]] \\
= \ & H(\pi) + \sum_{\rho} p_0(y_\rho; x_\rho) G(\rho) \\
= \ & H(\pi) + G(\pi).
\end{aligned}
$$

Thus, the theorem holds.

# B    Proof of Theorem 4

To prove Theorem 4, we first reduce probabilistic hypotheses (or mappings) to deterministic (or noiseless) ones by expanding the hypothesis space. Then, we apply a known result on deterministic hypotheses to obtain the result for the probabilistic hypotheses.

## B.1    An Equivalence between Probabilistic and Deterministic Hypotheses

First, we establish a relationship between probabilistic and deterministic hypotheses. Recall that $h \in \mathcal{H}$ is a probabilistic hypothesis, and $\mathbb{P}[h(x) = y|h] \in [0, 1]$ for all $h$ when $h$ itself is probabilistic. Let $T$ be a set of examples (without the labels) and let $y_T$ be the labeling of $T$. Let $\mathcal{D} = (T, y_T)$. Let $p_0$ be the prior on $\mathcal{H}$. The posterior $p_{\mathcal{D}}$ is obtained from $p_0$ using Bayes rule

$$p_{\mathcal{D}}[h] = p_0[h|\mathcal{D}] = \frac{p_0[h] \, \mathbb{P}[h(T) = y_T|h]}{p_0[h(T) = y_T]}.$$

From this noisy model for probabilistic hypothesis $h$, we construct an equivalent noiseless and deterministic one. We consider a hypothesis space $\mathcal{H}'$ such that $\mathcal{H}' = \{h'_{\mathbf{y}}\}_{\mathbf{y} \in \mathcal{Y}^{|X|}}$ and $h'_{\mathbf{y}}(x) = \mathbf{y}_{\langle\{x\}\rangle}$ for all $x \in X$. In this definition, for any $S \subseteq X$, $\mathcal{Y}^{|S|}$ is the set of all labelings of $S$ and $\mathbf{y}_{\langle S \rangle}$ is the projection of $\mathbf{y}$ on $S$, i.e. the labeling of $S$ according to $\mathbf{y}$. Hence, $\mathbf{y}_{\langle\{x\}\rangle}$ is the label of $x$ according to $\mathbf{y}$.

In the above definition, $\mathcal{H}'$ is indexed by the labelings of the pool $X$ and each $h'_{\mathbf{y}}$ in $\mathcal{H}'$ is a deterministic hypothesis. Further, we construct a prior $p'_0$ over $\mathcal{H}'$ such that $p'_0[h'_{\mathbf{y}}] = p_0[h(X) = \mathbf{y}] = \sum_{h \in \mathcal{H}} p_0[h] \, \mathbb{P}[h(X) = \mathbf{y}|h]$. The result is that $p'_0[h'_{\mathbf{y}}]$ is the probability that the labeling of $X$ is $\mathbf{y}$ in the probabilistic model. Given $\mathcal{D}$, the posterior $p'_{\mathcal{D}}$ on $\mathcal{H}'$ is obtained from $p'_0$ by

$$p'_{\mathcal{D}}[h'_{\mathbf{y}}] = \frac{p'_0[h'_{\mathbf{y}}] \, \mathbf{1}(\mathbf{y}_{\langle T \rangle} = y_T)}{\sum_{\mathbf{y} \in \mathcal{Y}^{|X|}} p'_0[h'_{\mathbf{y}}] \, \mathbf{1}(\mathbf{y}_{\langle T \rangle} = y_T)},$$

where $\mathbf{1}(A)$ is the indicator function for the event $A$. In essence, we have "moved" uncertainty associated with the likelihood $\mathbb{P}[h(T) = y_T|h]$ into the prior $p'_0$.

We now prove that the above two models are in fact equivalent in the sense that $p_{\mathcal{D}}[h(S) = y_S] = p'_{\mathcal{D}}[h'(S) = y_S]$ for any $S \subseteq X \setminus T$ and $y_S \in \mathcal{Y}^{|S|}$. This means that both models always give the same probability for the event $h(S) = y_S$. To prove this result, we need the following lemma about $p_0[\mathcal{D}] = p_0[h(T) = y_T]$.

**Lemma 1.** *We have* $p_0[h(T) = y_T] = \sum_{\mathbf{y} \in \mathcal{Y}^{|X|}} p'_0[h'_{\mathbf{y}}] \, \mathbf{1}(\mathbf{y}_{\langle T \rangle} = y_T).$

*Proof.* For a probabilistic hypothesis $h$, $p_0[h(T) = y_T] = \sum_{h \in \mathcal{H}} p_0[h] \, \mathbb{P}[h(T) = y_T|h]$. Expanding $\mathbb{P}[h(T) = y_T|h]$ by summing over all possible labelings of the remaining unlabeled examples in $X \setminus T$, we have

$$p_0[h(T) = y_T] = \sum_{h \in \mathcal{H}} p_0[h] \sum_{\mathbf{y} \in \mathcal{Y}^{|X|}} \mathbb{P}[h(X) = \mathbf{y}|h] \, \mathbf{1}(\mathbf{y}_{\langle T \rangle} = y_T)$$

$$= \sum_{\mathbf{y} \in \mathcal{Y}^{|X|}} \mathbf{1}(\mathbf{y}_{\langle T \rangle} = y_T) \sum_{h \in \mathcal{H}} p_0[h] \, \mathbb{P}[h(X) = \mathbf{y}|h]$$

$$= \sum_{\mathbf{y} \in \mathcal{Y}^{|X|}} \mathbf{1}(\mathbf{y}_{\langle T \rangle} = y_T) \, p'_0[h'_{\mathbf{y}}].$$

$\square$

Using Lemma 1, we can prove the following equivalence.

**Lemma 2.** *Let* $p_{\mathcal{D}}$ *and* $p'_{\mathcal{D}}$ *be the posteriors of the probabilistic and deterministic models respectively after observing the labeled examples* $\mathcal{D} = (T, y_T)$. *For any* $S \subseteq X \setminus T$ *and* $y_S \in \mathcal{Y}^{|S|}$, *we have* $p_{\mathcal{D}}[h(S) = y_S] = p'_{\mathcal{D}}[h'(S) = y_S].$

*Proof.* For the probabilistic hypotheses, we have

$$p_{\mathcal{D}}[h(S) = y_S] = \sum_{h\in\mathcal{H}} p_{\mathcal{D}}[h]\,\mathbb{P}[h(S)=y_S|h] = \sum_{h\in\mathcal{H}} \frac{p_0[h]\,\mathbb{P}[h(T)=y_T|h]}{p_0[h(T)=y_T]}\,\mathbb{P}[h(S)=y_S|h].$$

Expanding $\mathbb{P}[h(T)=y_T|h]\,\mathbb{P}[h(S)=y_S|h]$ by summing over all possible labelings of the remaining unlabeled examples in $X\setminus(T\cup S)$, we have

$$p_{\mathcal{D}}[h(S)=y_S] = \sum_{h\in\mathcal{H}} \frac{p_0[h]}{p_0[h(T)=y_T]} \sum_{\mathbf{y}\in\mathcal{Y}^{|X|}} \mathbb{P}[h(X)=\mathbf{y}|h]\,\mathbf{1}(\mathbf{y}_{\langle T\rangle}=y_T)\,\mathbf{1}(\mathbf{y}_{\langle S\rangle}=y_S)$$

$$= \sum_{\mathbf{y}\in\mathcal{Y}^{|X|}} \frac{\mathbf{1}(\mathbf{y}_{\langle T\rangle}=y_T)\,\mathbf{1}(\mathbf{y}_{\langle S\rangle}=y_S)}{p_0[h(T)=y_T]} \sum_{h\in\mathcal{H}} p_0[h]\,\mathbb{P}[h(X)=\mathbf{y}|h]$$

$$= \sum_{\mathbf{y}\in\mathcal{Y}^{|X|}} \frac{\mathbf{1}(\mathbf{y}_{\langle T\rangle}=y_T)\,\mathbf{1}(\mathbf{y}_{\langle S\rangle}=y_S)}{p_0[h(T)=y_T]} p_0'[h_{\mathbf{y}}'].$$

The last equality is from the definition of $p_0'[h_{\mathbf{y}}']$. From Lemma 1 and the definition of $p_{\mathcal{D}}'[h_{\mathbf{y}}']$:

$$\frac{p_0'[h_{\mathbf{y}}']\,\mathbf{1}(\mathbf{y}_{\langle T\rangle}=y_T)}{p_0[h(T)=y_T]} = \frac{p_0'[h_{\mathbf{y}}']\,\mathbf{1}(\mathbf{y}_{\langle T\rangle}=y_T)}{\sum_{\mathbf{y}\in\mathcal{Y}^{|X|}} p_0'[h_{\mathbf{y}}']\,\mathbf{1}(\mathbf{y}_{\langle T\rangle}=y_T)} = p_{\mathcal{D}}'[h_{\mathbf{y}}'].$$

Thus, $p_{\mathcal{D}}[h(S)=y_S] = \sum_{\mathbf{y}\in\mathcal{Y}^{|X|}} p_{\mathcal{D}}'[h_{\mathbf{y}}']\,\mathbf{1}(\mathbf{y}_{\langle S\rangle}=y_S) = p_{\mathcal{D}}'[h'(S)=y_S].$  □

## B.2 Near-optimality of the Noiseless Model

We now focus on the space $\mathcal{H}'$ of deterministic hypotheses. We will make use of the notations for the noiseless model in [1]. In this model, for a set of unlabeled examples $S\subseteq X$ and a hypothesis $h\in\mathcal{H}'$, we can define the version space $V(S,h)$ as the set of all hypotheses in $\mathcal{H}'$ that are consistent with $h$ on $S$. Formally, $V(S,h) = \{h'\in\mathcal{H}' : h'(S)=h(S)\}$. The probability of the version space $V(S,h)$ with respect to the prior $p_0'$ is

$$p_0'[V(S,h)] = \sum_{h'\in V(S,h)} p_0'[h'] = \mathbb{P}_{h'\sim p_0'}[h'(S)=h(S)\,|\,h].$$

Let $f(S,h) = 1 - p_0'[V(S,h)]$ be the version space reduction function. It is known that in the noiseless model, the version space reduction function $f(S,h)$ is adaptive monotone submodular [1]. Thus, the greedy adaptive policy selecting $x^* = \arg\max_x \mathbb{E}_{h\sim p_{\mathcal{D}}'}[f(S\cup\{x\},h)-f(S,h)]$, where $S$ is the previously selected set and $p_{\mathcal{D}}'$ is the current posterior of the noiseless model, is near-optimal. This property is stated in Theorem A below and is a direct consequence of Theorem 5.2 in [1].

**Theorem A.** *For any $k\geq 1$, in the noiseless model, let $\pi$ be the greedy adaptive policy that selects $k$ examples by the criterion $x^* = \arg\max_x \mathbb{E}_{h\sim p_{\mathcal{D}}'}[f(S\cup\{x\},h)-f(S,h)]$, where $S$ is the previously selected set and $p_{\mathcal{D}}'$ is the posterior after observing the labels of $S$. Let $\pi^*$ be the adaptive policy that selects the optimal $k$ examples in terms of the version space reduction objective. We have*

$$\mathbb{E}_{h_{\mathbf{y}}'\sim p_0'}[f(x_{\rho^{\pi,\mathbf{y}}},h_{\mathbf{y}}')] > \left(1-\frac{1}{e}\right)\mathbb{E}_{h_{\mathbf{y}}'\sim p_0'}[f(x_{\rho^{\pi^*,\mathbf{y}}},h_{\mathbf{y}}')],$$

*where $\mathbb{E}_{h_{\mathbf{y}}'\sim p_0'}[\cdot]$ is with respect to the distribution $p_0'[h_{\mathbf{y}}']$ and $x_{\rho^{\pi,\mathbf{y}}}$ is the set of unlabeled examples selected by $\pi$ (along the path $\rho^{\pi,\mathbf{y}}$) assuming the true labeling of $X$ is $\mathbf{y}$.*

Note that once we assume the true labeling of $X$ to be a fixed $\mathbf{y}$, the policy $\pi$ follows exactly one path from the root to a leave in the policy tree of $\pi$. This path is denoted by $\rho^{\pi,\mathbf{y}}$ in Theorem A. Using Theorem A and Lemma 2, we can now prove Theorem 4.

## B.3 Proof of Theorem 4

For any algorithm $\pi$, we have

$$\mathbb{E}_{h_{\mathbf{y}}'\sim p_0'}[f(x_{\rho^{\pi,\mathbf{y}}},h_{\mathbf{y}}')] = \sum_{\mathbf{y}} p_0'[h_{\mathbf{y}}']\left(1 - p_0'[V(x_{\rho^{\pi,\mathbf{y}}},h_{\mathbf{y}}')]\right)$$

$$= \sum_{\mathbf{y}} p_0'[h_{\mathbf{y}}']\left(1 - p_0'[h'(x_{\rho^{\pi,\mathbf{y}}}) = \mathbf{y}_{\langle x_{\rho^{\pi,\mathbf{y}}}\rangle}]\right).$$

By definition of $p_0'[h_{\mathbf{y}}']$, we have $p_0'[h_{\mathbf{y}}'] = p_0[h(X) = \mathbf{y}] = p_0[\mathbf{y}; X]$. From Lemma 2, $p_0'[h'(x_{\rho^\pi, \mathbf{y}}) = \mathbf{y}_{\langle x_{\rho^\pi, \mathbf{y}} \rangle}] = p_0[h(x_{\rho^\pi, \mathbf{y}}) = \mathbf{y}_{\langle x_{\rho^\pi, \mathbf{y}} \rangle}]$. Thus,

$$
\begin{aligned}
\mathbb{E}_{h_{\mathbf{y}}' \sim p_0'}[f(x_{\rho^\pi, \mathbf{y}}, h_{\mathbf{y}}')] &= \sum_{\mathbf{y}} p_0[\mathbf{y}; X] \left( 1 - p_0[h(x_{\rho^\pi, \mathbf{y}}) = \mathbf{y}_{\langle x_{\rho^\pi, \mathbf{y}} \rangle}] \right) \\
&= \sum_{\rho} \sum_{\mathbf{y}: \rho^\pi, \mathbf{y} = \rho} p_0[\mathbf{y}; X] \left( 1 - p_0[h(x_{\rho^\pi, \mathbf{y}}) = \mathbf{y}_{\langle x_{\rho^\pi, \mathbf{y}} \rangle}] \right) \\
&= \sum_{\rho} (1 - p_0[h(x_\rho) = y_\rho]) \sum_{\mathbf{y}: \rho^\pi, \mathbf{y} = \rho} p_0[\mathbf{y}; X] \\
&= \sum_{\rho} (1 - p_0^\pi[\rho]) \, p_0^\pi[\rho] \\
&= V(\pi).
\end{aligned}
$$

Hence, the inequality in Theorem A is equivalent to $V(\pi) > (1 - 1/e)V(\pi^*)$.

Thus, to prove Theorem 4, what remains is to prove that the example $x^*$ selected by $\pi^{\mathrm{maxGEC}}$ using Equation (3) satisfies $x^* = \arg\max_x \mathbb{E}_{h \sim p_{\mathcal{D}}'}[f(S \cup \{x\}, h) - f(S, h)]$.

In the deterministic (noiseless) case, for any $x \in X$, consider

$$
\begin{aligned}
\mathbb{E}_{h' \sim p_{\mathcal{D}}'} \left[ p_0'[V(S \cup \{x\}, h')] \right] &= \sum_{h' \in \mathcal{H}': p_{\mathcal{D}}'[h'] > 0} p_{\mathcal{D}}'[h'] \, p_0'[V(S \cup \{x\}, h')] \\
&= \sum_{y \in \mathcal{Y}} \sum_{h' \in \mathcal{H}': p_{\mathcal{D}}'[h'] > 0 \wedge h'(x) = y} p_{\mathcal{D}}'[h'] \, p_0'[V(S \cup \{x\}, h')].
\end{aligned}
$$

For all $h'$ satisfying $p_{\mathcal{D}}'[h'] > 0$, we have $p_{\mathcal{D}}'[h'] = \dfrac{p_0'[h']}{\sum_{h': p_{\mathcal{D}}'[h'] > 0} p_0'[h']}$.

Thus, if $h'$ also satisfies $h'(x) = y$, we have

$$
\begin{aligned}
p_0'[V(S \cup \{x\}, h')] &= \sum_{h': p_{\mathcal{D}}'[h'] > 0 \wedge h'(x) = y} p_0'[h'] \\
&= \sum_{h': p_{\mathcal{D}}'[h'] > 0 \wedge h'(x) = y} \left( p_{\mathcal{D}}'[h'] \sum_{h': p_{\mathcal{D}}'[h'] > 0} p_0'[h'] \right).
\end{aligned}
$$

Hence,

$$
\begin{aligned}
&\mathbb{E}_{h' \sim p_{\mathcal{D}}'} \left[ p_0'[V(S \cup \{x\}, h')] \right] \\
&= \sum_{y \in \mathcal{Y}} \sum_{h': p_{\mathcal{D}}'[h'] > 0 \wedge h'(x) = y} \left( p_{\mathcal{D}}'[h'] \sum_{h': p_{\mathcal{D}}'[h'] > 0 \wedge h'(x) = y} \left( p_{\mathcal{D}}'[h'] \sum_{h': p_{\mathcal{D}}'[h'] > 0} p_0'[h'] \right) \right) \\
&= \left( \sum_{h': p_{\mathcal{D}}'[h'] > 0} p_0'[h'] \right) \left( \sum_{y \in \mathcal{Y}} \sum_{h': p_{\mathcal{D}}'[h'] > 0 \wedge h'(x) = y} \left( p_{\mathcal{D}}'[h'] \sum_{h': p_{\mathcal{D}}'[h'] > 0 \wedge h'(x) = y} p_{\mathcal{D}}'[h'] \right) \right) \\
&= \left( \sum_{h': p_{\mathcal{D}}'[h'] > 0} p_0'[h'] \right) \left( \sum_{y \in \mathcal{Y}} \left( \sum_{h': p_{\mathcal{D}}'[h'] > 0 \wedge h'(x) = y} p_{\mathcal{D}}'[h'] \right)^2 \right) \\
&= \left( \sum_{h': p_{\mathcal{D}}'[h'] > 0} p_0'[h'] \right) \left( \sum_{y \in \mathcal{Y}} (p_{\mathcal{D}}'[h'(x) = y])^2 \right).
\end{aligned}
$$

Thus,

$$\arg\max_x \left\{ 1 - \sum_{y \in \mathcal{Y}} (p_{\mathcal{D}}'[h'(x) = y])^2 \right\} = \arg\min_x \sum_{y \in \mathcal{Y}} (p_{\mathcal{D}}'[h'(x) = y])^2$$

$$= \arg\min_x \mathbb{E}_{h' \sim p_{\mathcal{D}}'} [p_0'[V(S \cup \{x\}, h')]]$$

$$= \arg\max_x \mathbb{E}_{h' \sim p_{\mathcal{D}}'} [f(S \cup \{x\}, h')]$$

$$= \arg\max_x \mathbb{E}_{h' \sim p_{\mathcal{D}}'} [f(S \cup \{x\}, h') - f(S, h')].$$

Furthermore, by Lemma 2, the example $x^*$ selected by Equation (3) satisfies

$$x^* = \arg\max_x \left\{ 1 - \sum_{y \in \mathcal{Y}} (p_{\mathcal{D}}[h(x) = y])^2 \right\} = \arg\max_x \left\{ 1 - \sum_{y \in \mathcal{Y}} (p_{\mathcal{D}}'[h'(x) = y])^2 \right\}.$$

Thus, $x^* = \arg\max_x \mathbb{E}_{h' \sim p_{\mathcal{D}}'} [f(S \cup \{x\}, h') - f(S, h')]$ and Theorem 4 holds.

## C   Proof of Theorem 5

We use the same notations as in Section 3.1 in the main paper. In each iteration of Algorithm 1, the example $x^*$ selected for the current batch by Equation (4) satisfies

$$x^* = \arg\max_x \epsilon_g^p(S \cup \{x\}) = \arg\max_x \left\{ \epsilon_g^p(S \cup \{x\}) - \epsilon_g^p(S) \right\},$$

where $p$ is the current posterior in the probabilistic model. From Theorem 3, the batch $S$ selected in each iteration of Algorithm 1 is near optimal, i.e, it satisfies $\epsilon_g^p(S) > (1 - 1/e) \max_{S':|S'|=s} \epsilon_g^p(S')$. To prove the near-optimality for the whole batch algorithm, we can employ the same noiseless model $\mathcal{H}'$ as in Section B.1. From Lemma 2, $\epsilon_g^p(S) = 1 - \sum_{y_S} p[y_S; S]^2 = 1 - \sum_{y_S} p'[y_S; S]^2$, where $p'$ is the corresponding posterior in the noiseless model and the summations are over all possible labelings $y_S$ of $S$. The following proposition states that $1 - \sum_{y_S} p'[y_S; S]^2$ is equal to the expected version space reduction in the noiseless model.

**Proposition 1.** *For any $S \subseteq X$, in the noiseless model,*

$$\mathbb{E}_{h' \sim p'}[1 - p'[V(S, h')]] = 1 - \sum_{y_S} p'[y_S; S]^2.$$

*Proof.* In the noiseless model, we have $\mathbb{E}_{h'_{\mathbf{y}} \sim p'}[1 - p'[V(S, h'_{\mathbf{y}})]] = \mathbb{E}_{\mathbf{y} \sim p'}[1 - p'[V(S, h'_{\mathbf{y}})]]$, where the second expectation is with respect to $p'[\mathbf{y}; X] = p'[h'_{\mathbf{y}}]$. Furthermore,

$$\mathbb{E}_{\mathbf{y} \sim p'}[1 - p'[V(S, h'_{\mathbf{y}})]] = \mathbb{E}_{\mathbf{y} \sim p'}[1 - p'[\mathbf{y}_{\langle S \rangle}; S]] = \mathbb{E}_{y_S \sim p'}[1 - p'[y_S; S]],$$

where $\mathbb{E}_{y_S \sim p'}[\,\cdot\,]$ is the expectation with respect to the distribution $p'[\,\cdot\,; S]$. Hence,

$$\mathbb{E}_{h'_{\mathbf{y}} \sim p'}[1 - p'[V(S, h'_{\mathbf{y}})]] = \mathbb{E}_{y_S \sim p'}[1 - p'[y_S; S]] = 1 - \sum_{y_S} p'[y_S; S]^2.$$

$\square$

Thus, $\epsilon_g^p(S)$ is equivalent to the expected version space reduction in the noiseless model with deterministic hypotheses. So, in the noiseless model, Algorithm 1 is equivalent to the BatchGreedy algorithm proposed in [2]. According to the results in [2], the version space reduction after observing the labeling of each batch is monotone adaptive submodular. Furthermore, from Theorem 3, the average version space reduction after selecting each batch is near-optimal, i.e, each iteration of Algorithm 1 is an $e/(e-1)$-approximate greedy step [1].

For any $k \geq 1$, let $\pi_b^{\text{maxGEC}}$ be the policy selecting $k$ batches using the batch maxGEC policy and $\pi_b^*$ be the batch policy selecting the optimal $k$ batches with respect to the policy Gibbs error objective. From Theorem 5.2 in [1],

$$\mathbb{E}_{h'_{\mathbf{y}} \sim p_0'}[1 - p_0'[V(x_{\rho^{\pi_b^{\text{maxGEC}},\mathbf{y}}}, h'_{\mathbf{y}})]] \geq (1 - e^{-(e-1)/e})\mathbb{E}_{h'_{\mathbf{y}} \sim p_0'}[1 - p_0'[V(x_{\rho^{\pi_b^*,\mathbf{y}}}, h'_{\mathbf{y}})]],$$

where $p'_0$ is the prior of the noiseless model and $x_{\rho^{\pi_b}, \mathbf{y}}$ is the set of all examples selected by the batch algorithm $\pi_b$ after $k$ iterations ($k\,s$ examples in total), assuming the true labeling of the pool $X$ is $\mathbf{y}$.

From Section B.3, $\mathbb{E}_{h'_{\mathbf{y}} \sim p'_0}[1 - p'_0[V(x_{\rho^{\pi_b}, \mathbf{y}}, h'_{\mathbf{y}})]] = V(\pi_b)$ for any policy $\pi_b$. Thus, we obtain Theorem 5.

## D   Derivation for the Approximation of Gibbs Error in Bayesian CRFs

We have:

$$
\sum_{\vec{y}} (p_{\mathcal{D}}[\vec{y}; \vec{x}])^2 \approx \sum_{\vec{y}} \left( \frac{1}{N} \sum_{j=1}^{N} P_{\lambda^j}[\vec{y}|\vec{x}] \right)^2
$$

$$
= \frac{1}{N^2} \sum_{\vec{y}} \left( \sum_{j=1}^{N} \frac{\exp\left( \sum_{i=1}^{m} \lambda_i^j F_i(\vec{y}, \vec{x}) \right)}{Z_{\lambda^j}(\vec{x})} \right)^2
$$

$$
= \frac{1}{N^2} \sum_{j=1}^{N} \sum_{t=1}^{N} \frac{1}{Z_{\lambda^j}(\vec{x}) Z_{\lambda^t}(\vec{x})} \sum_{\vec{y}} \exp\left( \sum_{i=1}^{m} \lambda_i^j F_i(\vec{y}, \vec{x}) \right) \exp\left( \sum_{i=1}^{m} \lambda_i^t F_i(\vec{y}, \vec{x}) \right)
$$

$$
= \frac{1}{N^2} \sum_{j=1}^{N} \sum_{t=1}^{N} \frac{1}{Z_{\lambda^j}(\vec{x}) Z_{\lambda^t}(\vec{x})} \sum_{\vec{y}} \exp\left( \sum_{i=1}^{m} (\lambda_i^j + \lambda_i^t) F_i(\vec{y}, \vec{x}) \right)
$$

$$
= \frac{1}{N^2} \sum_{j=1}^{N} \sum_{t=1}^{N} \frac{Z_{\lambda^j + \lambda^t}(\vec{x})}{Z_{\lambda^j}(\vec{x}) Z_{\lambda^t}(\vec{x})}.
$$

Thus, $\epsilon_g^{p_{\mathcal{D}}}(\vec{x}) = 1 - \sum_{\vec{y}} (p_{\mathcal{D}}[\vec{y}; \vec{x}])^2 \approx 1 - \frac{1}{N^2} \sum_{j=1}^{N} \sum_{t=1}^{N} \frac{Z_{\lambda^j + \lambda^t}(\vec{x})}{Z_{\lambda^j}(\vec{x}) Z_{\lambda^t}(\vec{x})}.$

## E   Experimental Results for Text Classification using Bayesian Transductive Naive Bayes with Batch Sizes $s = 20, 30$

Table 1: AUC of different learning algorithms with batch size $s = 20$.

| Task | TPass | maxGEC | LC | NPass | LogPass | LogFisher |
|---|---|---|---|---|---|---|
| alt.atheism/comp.graphics | 87.62 | 91.52 | 91.70 | 84.85 | 91.28 | **93.37** |
| talk.politics.guns/talk.politics.mideast | 84.23 | 92.52 | **92.56** | 80.61 | 85.89 | 86.93 |
| comp.sys.mac.hardware/comp.windows.x | 73.96 | **91.71** | 89.98 | 74.79 | 85.83 | 88.06 |
| rec.motorcycles/rec.sport.baseball | 93.65 | **95.95** | 95.93 | 92.04 | 89.25 | 93.11 |
| sci.crypt/sci.electronics | 61.10 | 86.19 | 85.97 | 61.28 | 82.80 | **86.93** |
| sci.space/soc.religion.christian | 92.44 | **95.77** | **95.77** | 89.67 | 91.04 | 93.48 |
| soc.religion.christian/talk.politics.guns | 91.11 | **94.56** | **94.56** | 85.41 | 90.09 | 93.12 |
| Average | 83.44 | **92.60** | 92.35 | 81.23 | 88.02 | 90.71 |

Table 2: AUC of different learning algorithms with batch size $s = 30$.

| Task | TPass | maxGEC | LC | NPass | LogPass | LogFisher |
|---|---|---|---|---|---|---|
| alt.atheism/comp.graphics | 87.72 | 92.22 | 92.22 | 85.27 | 91.05 | **92.88** |
| talk.politics.guns/talk.politics.mideast | 85.13 | **92.20** | 92.17 | 81.00 | 85.63 | 86.35 |
| comp.sys.mac.hardware/comp.windows.x | 72.81 | **88.58** | 88.53 | 74.53 | 85.75 | 87.52 |
| rec.motorcycles/rec.sport.baseball | 94.03 | 96.21 | **96.22** | 92.09 | 89.03 | 92.22 |
| sci.crypt/sci.electronics | 61.71 | 86.12 | 85.25 | 61.62 | 82.74 | **86.31** |
| sci.space/soc.religion.christian | 91.09 | **95.86** | **95.86** | 88.76 | 90.88 | 92.82 |
| soc.religion.christian/talk.politics.guns | 91.00 | **95.54** | **95.54** | 85.19 | 89.65 | 91.89 |
| Average | 83.36 | **92.39** | 92.26 | 81.21 | 87.82 | 90.00 |