[Reviews · NeurIPS 2013]

Submitted by Assigned_Reviewer_4

Overview:
The authors propose the Gibbs error criterion for active learning; seeking the samples that maximize the expected Gibbs error under the current posterior. They propose a greedy algorithm that maximises this criterion (Max-GEC). The objective reduces to maximising a specific instance of the Tsallis entropy of the predictive distribution which is very similar to Maximum Entropy Sampling (MES) which uses the Shannon entropy of the predictive distribution. They consider the non-adaptive, adaptive and batch settings separately, and in each setting they prove using submodularity results that the greedy approach achieves near-maximal performance compared to optimal policy. They show how to implement their fully adaptive policy (approximately) in CRFs with application to named entity recognition, and implement the batch algorithm with a Naive Bayes classifier, with application to a text classification task.

Quality:
Their algorithm appears to be a sensible approach, with the Gibbs error being very closely related to the Bayes error. Although it appears very similar to a number of approaches in the literature (MES, Query by Committee), it exhibits some useful theoretical and practical properties. Primarily, they are able to show near-optimal bounds in the adaptive greedy setting with probabilistic predictions; previous approaches have been shown only to be optimal in the non-adaptive or noiseless case. A second advantage is that their objective permits a simpler computation of their objective with CRFs when approximating integrals over the posterior with samples; it would be interesting to see a discussion of whether this rearrangement is extensible to other models?

However, I have a concern about the practicality of the algorithm. In particular in the non-adaptive/batch case, a sum over the product space of all possible labellings of the batch (S) is required (Eqns. 2, 4). When applying batch Max-GEC to the NB classifier they approximate the sum using Gibbs sampling. Given how large the space of possible labelings of the batch could be, this may require a very large number of samples to get a reasonable estimate, the requirement to compute/estimate this sum seems to restrict the batch algorithm to either small batches or models in which computing predictions is cheap.

The experiment section is fairly convincing; using two tasks and a number of different datasets, they show max-GEC usually outperforms a number of baselines, although the improvement over LeastConf seems marginal at best. A concern I have is that for the CRF model, SegEnt/LeastConf produce almost as good results as Max-GEC, which is perhaps unsurprising given the similarity between the algorithms, however, for SegEnt/LeastConf the authors use only the MAP hypothesis to compute uncertainties, and for Max-GEC they show improvement by integrating over parameters (using samples). They should compare also to SegEnt and LeastConf with parameter integration, these approaches may be more sensitive to accurate estimation of uncertainties and I am unconvinced that there would necessarily be a performance difference after accounting for parameter uncertainty.

Clarity:
The paper is largely clearly written, however, although they do define the notation, I find some of the choices of notation somewhat confusing. In particular use of a generic p to denote the posterior as opposed to the prior p_0 is a bit unclear (perhaps it could be subscripted by the data used to train the model). Also the use of the symbol traditionally used for ‘conditioning’ (as in y_{A|X}) to denote the labelling of X according to A makes the paper harder to read. The use of E(\rho) to denote the set of unlabelled examples looks a lot like an expectation, also regarding expectations it would be useful to subscript them with the distribution over which the expectation is being taken e.g. for an expectation under the prior E_y -> E_{p_0(y)}.

Significance:
Although the optimality proofs provide an interesting insight into this particular criterion, and provide a decent theoretical contribution, the algorithm itself is sufficiently similar to a number of proposed approaches and so this paper does not represent a very significant practical contribution to the field of active learning.

Summary: Maximising Gibbs error is intuitively a sensible approach for active learning, and the optimality guarantees presented in the paper verify this, furthermore the experiments with CRFs and NB classifiers show reasonable performance. However, the algorithm itself is very similar to a large number of proposed approaches based upon version space reduction/entropy sampling, I also have some concerns about the practicality/extensibility of the batch algorithm due to the requirement to compute an exponentially hard sum.

Submitted by Assigned_Reviewer_5

The authors consider an active learning policy of choosing examples
with the highest Gibbs error. They consider three setting (1)
non-adaptive, (2) adaptive, and (3) batch. The implication is that by
choosing those examples with highest Gibbs error, the overall error
probability will be minimized. While this has intuitive appeal, I am
unaware of any formal proofs that show this to be the case. Certainly,
SVMs rely on a similar strategy (and there are formal proofs). In any
event, a reference to any analysis on this choice would greatly
improve the paper.

Regarding the non-adaptive policy, the authors state that this is
equivalent to selecting examples prior to labeling. However, the
greedy selection of equation (2) at least implicitly includes the
labels. Otherwise, how is it possible to compute the sequential
Tsallis entropy term?

In the adaptive policy the authors show (in supplementary) that the
greedy criterion satisfies an adaptive montone submodular property and
hence may exploit known bounds on such reward functions.

The batch setting (a bit misnamed) intersperses posterior updates with
greedy selections of small batches of data. It is a compromise between
the non-adaptive and adaptive approaches.

One strong criticism is that the notation is constantly being
redefined. For example, the Gibbs error of line 110 is defined adn
then apparently discarded in section 2.1 in favor of \epsilon_g (line
134), which is then discarded in favor of g_{p_0} (line 137). I can
appreciate trying to simplify the notation, but these changes make it
difficult to follow the agruments.

The authors propose a sampling approach for approximating the Gibbs
error in exponential models (e.g. CRFs) and batch Gibbs sampling for
Naive Bayes models. Experiments are performed on two tasks
(recognition and text classification) using the two models and
associated estimators of Gibbs error.

Provding the related work section at the end seems to be an odd choice.


line 108 y_u is described as the "true labeling" of the data. It later appears as a subscript in Eq. 1 implying that it is a random variable. Just after Eq. 1 "for a fixed labeling" also refers to y_u. Please clarify.

line 109 "For a prior p_o..." prior what? This same symbol is used to describe the posterior over mappings given data+labelings in line 098. They do not appear to be the same thing.

line 134 The steps from E_{y_s} to Tsallis entropy are not obvious (at least to me).

minor comments (i.e. no need to rebut):

Is the set of mappings finite or countably infinite. Please clarify...perhaps it doesn't matter.

line 089 using p[h] for p_0[h|D] is a bit distracting as the same notation is used to denote a marginal event probability in the very next sentence.



Summary: The authors exploit submodular properties of Gibbs error and its relation to Tsallis entropy to establish guarantees for greedy methods for online learning.

Submitted by Assigned_Reviewer_7

This paper considers pool-based active learning and batch mode active learning using a greedy algorithm which selects examples to label to maximize a quantity called the policy Gibbs error. The proposed algorithms can be seen as generalizations of prior work on version-space reduction algorithms, and benefits from similar (constant-factor) approximation guarantees (based on the adaptive submodularity of the policy Gibbs error).

The method is flexible, and can be used whenever the policy Gibbs error can be computed in practice. The authors evaluate their algorithm with two applications -- entity identification with CRFs and Bayesian transductive naive Bayes models -- with modest improvements on prior work. Overall this is a nice paper in an important area.

The exposition is good, though section 2 could stand to be improved.
A sentence defining the Gibbs classifer would be nice.

The authors claim their work does not require an explicit noise model, in contrast to earlier work. It would be nice to point out what noise models their methods can handle (i.e., that the noise model is implicit in the probabilistic model and is limited by computational concerns).

There appear to be some interesting connections between the Tsallis entropy and the progress measures in prior work (where terms like 1 - sum_i p_i^2 often appear).
Summary: See the second paragraph above.
Author Feedback

Author rebuttal: Review 1

1. Models other than CRF
We have discussed a conditional model in section 3.1 and a generative model in section 3.2. The conditional model in 3.1 is the exponential model, and hence it covers a wide class of models, including linear chain CRF, semi-Markov CRF and sparse higher-order semi-Markov CRF. As can be seen in the last set of equations in 3.1, computing our criterion basically consists of (1) the partition function, and (2) sampling of \lambda. If we further use the MAP estimate of \lambda, then only (1) is needed. Hence our criterion is applicable to classes of models for which the partition function can be computed efficiently, e.g., tree CRF.

2. Large number of samples for large batch sizes and restriction to small batch sizes
We agree this is a limitation of the batch algorithm. However, in some real problems, we may prefer small batches to large ones. For example, if we have a small number of annotators and labeling one example takes a long time, we may want to select a batch size that matches the number of annotators. In this case, the annotators can label the examples concurrently while we can make use of the labeled examples as soon as they are available. If we choose a large batch, it would take a long time to label the whole batch and we cannot use the labeled examples until all the examples in the batch are labeled.

3. Different bounding constant in Theorem 3
The batch algorithm has a different bounding constant because it uses two levels of approximation to compute the batch policy: At each iteration, it approximates the optimal batch by greedily choosing one example at a time using equation 4 (1st approximation). Then it uses these chosen batches to approximate the optimal batch policy (2nd approximation). In the fully adaptive and non-adaptive approaches, we only need to make one approximation. In the fully adaptive case, the batch size is 1, so we can always choose the optimal batch at each iteration. Thus, we only need the 2nd approximation. In the non-adaptive case, we only choose 1 batch. So, we only need the 1st approximation.

4. Parameter integration for SegEnt/LeastConf (using samples)
In Bayesian CRF, although we can sample a finite set of models from the posterior, as far as we know, there is currently no simple or efficient way to compute the SegEnt/LeastConf criteria from the samples, except for using only the MAP estimation. The main difficulty is to compute a summation (minimization for the LeastConf criterion) over all the outputs y's in the complex structured models. For max-GEC, the summation can be rearranged to obtain the partition functions, but we cannot do so for SegEnt/LeastConf. This is an advantage of using max-GEC since it can be computed efficiently from the samples using known inference algorithms.

Review 2

1. Proof that our criterion minimizes the overall error probability
We are also unaware of any formal proof of this. However, in Bayesian setting, the Gibbs error upper bounds the Bayes error, and hence serves as a motivation for us to investigate this criterion.

2. Computing the sequential Tsallis entropy in non-adaptive policy
In equation 2, we do not know the true labeling of S_i. So, we take a summation over all the possible labelings of S_i \cup {x}. For example, if there are two labels, this summation is over 2^{|S_i|+1} labelings. In other words, we compute the Tsallis entropy for the distribution p_0[y_{S_i \cup {x}}; S_i \cup {x}] over these 2^{|S_i|+1} labelings. Since the support of this distribution (the number of labelings) is exponentially large, we can approximate the Tsallis entropy using Algorithm 2.

3. On redefinition of Gibbs error
The Gibbs error of line 110 is different from the definition at line 134/137. The error of line 110 is the error along one path of the policy tree, while the error at line 134/137 is the average error of the whole (non-adaptive) policy tree. We will improve the clarity of the notations.

4. Clarification for y_U at line 108
For the unlabeled examples, we do not know their true labeling. So, we can think of the true labeling as a random variable, whose probability is determined by the model. Given a model, which is any distribution on the hypothesis space (p_0 in this case), the probability of a labeling y_U can be computed using the equation at line 090 and is equal to p_0[y_U; U] (defined at line 105 for any distribution p).
If a fixed y_U is really the true labeling, we can compute the error of a Gibbs classifier on the set selected by a policy \pi (this set corresponds to exactly one path from the root to a leaf of the policy tree of \pi). This error is defined as \mathcal{E}(...) at line 110. However, we do not know which y_U is really the true labeling. So, we have to take an expectation over all the possible y_U in the definition of the policy Gibbs error (equation 1).
The sentence below equation 1 is to explain the above fact that when a y_U is really the true labeling, the policy will select examples along exactly one path down the policy tree.

5. Clarification for p_0 at line 109
The prior p_0 is a probability distribution over the hypothesis space \mathcal{H} (line 088). For any probability distribution on the hypothesis space (including both the prior p_0 and the posterior p), we can induce a probability for any event A using the equation at line 090.

6. The steps from E_{y_s} to Tsallis entropy at line 134
Since E_{y_S}[.] is with respect to the distribution p_0[y_S;S], we have:
E_{y_S}[1 - p_0[y_S;S]] = 1 - E_{y_S}[p_0[y_S;S]] = 1 - \sum_{y_S}(p_0[y_S;S] * p_0[y_S;S]) = 1 - \sum_{y_S}(p_0[y_S;S])^2.
This is the Tsallis entropy for the distribution p_0[y_S;S] over all the possible labelings of S.

Review 3
We agree with most points from reviewer 3 and will make the suggested changes.

Notes: The notation has changed in the final paper to increase clarity.